# Partners’ Relationship Mindfulness Promotes Better Daily Relationship Behaviours for Insecurely Attached Individuals

**DOI:** 10.3390/ijerph17197267

**Published:** 2020-10-05

**Authors:** Taranah Gazder, Sarah C. E. Stanton

**Affiliations:** Department of Psychology, University of Edinburgh, Edinburgh EH8 9JZ, UK; sarah.stanton@ed.ac.uk

**Keywords:** attachment, mindfulness, dyadic data, longitudinal, relationships

## Abstract

Attachment anxiety and avoidance are generally associated with detrimental relationship processes, including more negative and fewer positive relationship behaviours. However, recent theoretical and empirical evidence has shown that positive factors can buffer insecure attachment. We hypothesised that *relationship mindfulness* (RM)—open or receptive attention to and awareness of what is taking place internally and externally in a current relationship—may promote better day-to-day behaviour for both anxious and avoidant individuals, as mindfulness improves awareness of automatic responses, emotion regulation, and empathy. In a dyadic daily experience study, we found that, while an individual’s *own* daily RM did not buffer the effects of their own insecure attachment on same-day relationship behaviours, their *partner’s* daily RM did, particularly for attachment avoidance. Our findings for next-day relationship behaviours, on the other hand, showed that lower (vs. higher) prior-day RM was associated with higher positive partner behaviours on the following day for avoidant individuals and those with anxious partners, showing this may be an attempt to “make up” for the previous day. These findings support the Attachment Security Enhancement Model and have implications for examining different forms of mindfulness over time and for mindfulness training.

## 1. Introduction

Attachment orientations play a vital role in shaping emotional patterns, relationship behaviours, and wellbeing [1,2]. Attachment orientations are patterns of expectations, needs, emotions, and social behaviour which result from a particular history of attachment experiences, usually beginning in relationships with caregivers in infancy [3]. Partners’ *attachment anxiety* (i.e., worries about rejection and abandonment) and *attachment avoidance* (i.e., discomfort with intimacy and desire for emotional independence) are typically associated with negative relationship outcomes such as lower general satisfaction, connectedness, and support, and higher destructive interaction and conflict [1,3]. However, in recent years, research has begun to discover how to buffer insecure attachment (a term used to describe individuals who score high on attachment anxiety or avoidance) and improve relationship outcomes [4,5,6]. Our understanding of the specific processes that buffer attachment insecurity day-to-day, however, is still in its infancy. We propose that a potential process through which more anxiously and avoidantly attached individuals might enjoy better relationships over time is *relationship mindfulness (RM)*, an open and receptive awareness to one’s partner and relationship in the present moment [7]. RM may buffer attachment insecurity through an increased awareness of automatic reactions and increased emotion regulation and empathy, thereby downregulating engagement in hyperactivating (excessive proximity-seeking) and deactivating (lack of proximity-seeking) strategies [7,8,9]. The goal of the present research was to investigate how day-to-day RM might alter anxiously and avoidantly attached individuals’ same-day and next-day positive and negative relationship behaviours in a longitudinal study of romantic couples.

### 1.1. Attachment Theory

A core tenet of attachment theory is that early interactions with caregivers predict cognition, affect, and behaviour across the lifespan [10,11,12]. During times of distress, securely attached individuals (who score low on anxiety and avoidance) respond by enacting relationship-maintaining behaviours (e.g., affiliation, caregiving) to re-establish a positive bond with their partner [3]. Securely attached individuals have positive working models of themselves and others; that is, they believe that close others will provide them with a secure base from which to explore and a safe haven in times of distress. However, there are important individual differences in attachment functioning [10]. Anxiety and avoidance are two distinct dimensions which comprise attachment insecurity [13], and stem from early experiences with inconsistent or unresponsive care in infancy. Individuals high in attachment anxiety or avoidance have learned through experience that security-based strategies (proximity-seeking and relationship-maintenance behaviours) are ineffective, and therefore engage in secondary attachment strategies which can have detrimental consequences for the relationship. Anxiously attached individuals have experienced inconsistent responsiveness from caregivers and hence engage in hyperactivating strategies, which involve intensified proximity-seeking behaviours to try to get their partner’s attention [3]. Meanwhile, avoidantly attached individuals have experienced caregivers who are unresponsive to their needs and hence engage in deactivating strategies, in which individuals give up proximity-seeking efforts without having their attachment needs fulfilled [3].

Anxious individuals enact hyperactivating strategies in an attempt at relationship maintenance [14]. Hyperactivating strategies, often triggered by perceived threat, involve excessive proximity seeking, rumination, and hypervigilance [3], which often manifest as negative relationship behaviours [15,16]. These strategies tend to intensify the magnitude of negative cognitions, emotions, and behaviours [1,17]. For instance, anxious individuals escalate conflict [15], engage in destructive behaviours during problem discussions [18] and negative behaviours during support interactions [19], and often attempt to induce guilt in their partners [16].

Anxiously attached individuals also appear to enact lower rates of positive behaviour, perhaps due to their preoccupation with the fulfilment of attachment needs. This leaves them with fewer resources to invest in other behavioural systems associated with positive relationship behaviours, such as caregiving and affiliation [3]. Empirical evidence suggests anxious individuals tend to engage in less constructive interaction, provide less support in their relationships [1], and engage in less relationship self-regulation [20].

Avoidantly attached individuals also enact more negative and fewer positive relationship behaviours, but for different reasons. Avoidantly attached individuals tend to experience indifference and employ deactivating strategies [14,17]. Avoidant individuals tend to enact more destructive behaviour in conflict [1,21] and exhibit hostile behavioural responses towards a distressed romantic partner [22], providing reason to believe that avoidance, too, would be associated with higher rates of negative behaviours.

As deactivating strategies involve establishing emotional distance from a partner, they often lead to fewer positive relationship behaviours [3,20]. The use of deactivating strategies blocks full activation of the sexual and affiliation systems, as well as prosocial behaviour. Avoidant individuals seek and provide less social support [23], self-disclose less, and are uncomfortable with a high self-disclosing partner [24], indicating that they experience, enjoy, and create positive interpersonal experiences less than individuals low in avoidance.

There is growing evidence that protective factors can buffer the typically negative associations between insecure attachment and healthy relationship outcomes. The Attachment Security Enhancement Model (ASEM) [4] proposes that when individuals tailor their responses to address their partner’s anxious or avoidant thoughts and feelings, the insecure partner and relationship benefit. Attachment anxiety can be buffered by utilising *safe* strategies, which involve conveying a strong, intimate emotional bond and providing reassurance while also deescalating negative emotions. Meanwhile, avoidance can be buffered by enacting *soft* strategies, behaviours sensitive to an avoidant person’s discomfort with emotional interactions. They may involve communicating how and why certain needs and requests are reasonable, respecting the avoidant individual’s need for autonomy within the context of positive relationship experiences. Recent empirical research has found that avoidance is buffered by gratitude, positive relationship experiences, and a partner’s non-verbal affectionate communication [6,25,26], whereas anxiety is buffered by physical touch, general mindfulness, and higher perceived partner support [5,27,28]. A process which may buffer both anxiety and avoidance is RM, as, theoretically, it should downregulate hyperactivating and deactivating strategies [27].

### 1.2. Mindfulness and Relationships

Mindfulness is broadly defined as an open or receptive attention to and awareness of what is taking place internally and externally in the present moment [29]. General mindfulness is associated with lower stress and mood disturbance [29,30] and greater empathy, relationship satisfaction, closeness, and more constructive responses to stress [31,32,33]. Karremans et al.’s [8] theoretical framework of mindfulness in relationships outlines how mindfulness may improve awareness of automatic responses, emotional regulation, and executive control, and increase self–other connectedness [8,29,33,34,35], which, in turn, may improve relationship outcomes.

Whereas securely attached individuals can adjust their cognitive structures to new information, individuals who are anxiously or avoidantly attached struggle to revise their negative working models of themselves and others, causing them to engage in maladaptive ways of coping, which often generate more negative and fewer positive interpersonal experiences [3]. Theoretically, mindfulness should downregulate hyperactivating or deactivating attachment tendencies [8,27]. Anxious individuals tend to resist information inconsistent with their preconceptions, whereas mindful people are in touch with their in-the-moment experiences. This reduced preoccupation and increased involvement in the present moment may help anxious individuals enact fewer negative and more positive relationship behaviours [27]. Indeed, studies have shown that mindfulness buffers anxiously attached individuals’ wellbeing [2] and promotes their relationship stability [27].

Mindfulness may also soften the hostility associated with attachment avoidance [17] and improve awareness of automatic cognitive, affective, and behavioural responses as well as emotion regulation [8], thereby creating more flexible working models. By buffering deactivating strategies, mindful attention and awareness may also lead avoidant individuals to engage with their relationships more constructively. Moreover, consistent with Karremans’s [8] model, mindfulness has also been associated with higher interpersonal closeness and relationship satisfaction [4,33], implying that mindfulness may foster factors such as intimacy, which avoidant individuals generally struggle with [1].

Research has primarily focused on the benefits of general mindfulness. However, RM (i.e., how mindful individuals are in their current relationship) is a unique form of mindfulness which predicts positive and negative relationship quality over and above general mindfulness [7,9]. RM thus contributes to positive relationship outcomes in a more targeted way. The literature applying RM to relationships over time, however, is nascent, and no research to date has examined the role of RM in buffering insecurely attached individuals’ day-to-day relationship behaviours. The daily diary methodology employed in the current research is particularly important because, while mindfulness is often measured as a trait, it fluctuates over time [8,36]. Therefore, assessing the impact of daily fluctuations in RM is more informative than the use of a trait measure.

Additionally, most previous work has examined actor effects (i.e., individuals’ *own* characteristics predicting their own outcomes). Partners in a relationship are interdependent; the cognitions, emotions, and behaviours of one partner can affect the outcomes of the other [37]. It is therefore important to take a dyadic perspective in examining the buffering effects of RM [8]. While the empirical evidence for partner effects (i.e., the *partner’s* characteristics predicting individuals’ own outcomes) of mindfulness is mixed [9,31], there is theoretical rationale for partner effects in the context of RM buffering attachment insecurity. In line with the ASEM [4], experiencing higher RM may help partners to be aware of their insecurely attached partner’s thoughts, feelings, and fears, making them more likely to enact the appropriate safe or soft strategies, thereby buffering the effects of insecure attachment on their partner’s, or their own, relationship behaviours [20]. Experiencing higher RM on a given day may also prevent one’s own attachment insecurity from undermining partner relationship behaviours. Research has found that one partner’s mindfulness lowers the other partner’s dominance and reactivity during relationship conflict [38] and negative affect [39]. Additionally, Kimmes et al. [9] found that actor RM predicted better partner relationship quality and general wellbeing. Thus, it is plausible that one partner’s mindfulness could buffer the effects of the other partner’s insecure attachment on either partner’s relationship behaviours.

### 1.3. Research Overview and Hypotheses

This research investigated how day-to-day RM may buffer more anxiously and avoidantly attached individuals’ tendency to enact fewer positive and more negative relationship behaviours on a given day. Additionally, we explored whether daily RM buffered relations between attachment anxiety and avoidance and next-day relationship behaviours to examine whether potential buffering effects held over a longer period. This is both theoretically and practically valuable because relationship behaviours are predictive of more global outcomes, including relationship quality, trust, and wellbeing, which have implications for public health [3,25,40,41]. We tested actor–actor, partner–partner, and cross-partner effects of attachment and daily RM predicting daily behaviours. The present research was preregistered at (https://osf.io/7hw5x/) [42].

**Confirmatory Hypothesis 1** **(CH1).**We predicted that higher (vs. lower) levels of attachment anxiety would be associated with higher daily levels of negative relationship behaviours and lower daily levels of positive relationship behaviours, consistent with theory and prior research [1,3,15,18,19] (**CH1a**). We also predicted that higher (vs. lower) levels of attachment avoidance would be associated with higher daily levels of negative relationship behaviours and lower daily levels of positive relationship behaviours, consistent with theory and previous studies (e.g., [1,3,21,22] (**CH1b**)).

**Confirmatory Hypothesis 2** **(CH2).**We predicted that higher (vs. lower) daily levels of RM would be associated with lower levels of daily negative behaviours and higher levels of daily positive behaviours, aligned with prior mindfulness research [7,31,33].

**Confirmatory Hypothesis 3** **(CH3).**We predicted that daily RM would moderate the link between attachment anxiety and relationship behaviours. Specifically, we predicted that daily RM would be associated with fewer negative behaviours and more positive behaviours for individuals both high and low in attachment anxiety, but that the slope would be steeper for individuals high in attachment anxiety. RM may decrease the rumination and hyperactivating strategies employed by anxious individuals, thereby decreasing negative relationship behaviours and helping anxious individuals devote more resources to engaging in positive behaviours [2,8,27] (**CH3a**).

Similarly, we predicted that daily RM would moderate the link between attachment avoidance and relationship behaviours. We predicted that daily RM would be associated with lower levels of negative behaviours and higher levels of positive behaviours for individuals both high and low in attachment avoidance, but that the slope would be steeper for individuals high in attachment avoidance. RM may increase avoidant individuals’ comfort with intimacy and improve awareness of automatic responses, helping it to buffer avoidance, as other protective factors do [6,8,25] (**CH3b**).

**Exploratory Hypothesis 1** **(EH1).**We explored whether RM buffers the relations between attachment insecurity and next-day positive and negative relationship behaviours to examine whether the above hypothesised effects held over a 24-h period.

**Exploratory Hypothesis 2** **(EH2).**While all our hypotheses concerned Actor × Actor interactions, we also explored Actor × Partner, Partner × Actor, and Partner × Partner interactions of attachment orientations and RM to examine their associations with positive and negative behaviours. It is possible that a similar buffering pattern to the actor predictions above may emerge when considering partner variables; however, to the best of our knowledge, there is no previous literature that leads to a strong *a priori* prediction that this will be the case. Thus, we kept analyses regarding partner effects exploratory.

## 2. Materials and Methods

### 2.1. Participants

One hundred romantic couples (87 heterosexual, 9 lesbian, 1 gay, and 3 other non-binary) were recruited from the University of Edinburgh and surrounding community via social media posts, advertisements in local magazines, and at local wedding fairs. Participants were 18–64 years of age (*M*_years_ = 24.15, *SD*_years_ = 6.61) and were in relationships lasting 3 months to 35.50 years (*M*_years_ = 2.84, *SD*_years_ = 4.41). Most of the sample identified their ethnicity as White (85.50%). Approximately 85.50% of participants were casually or exclusively dating their current partner, and 14.50% were common law, engaged, in a civil partnership, or married. A minority of couples (38.00%) were living together.

### 2.2. Measures and Procedure

Data were collected as part of a larger study of couples’ psychological experiences over time (see https://osf.io/ekv6x/) [43]. The study had three phases: an initial 2-h lab session (Phase 1), a 14-day diary period (Phase 2), and a follow-up survey two months later (Phase 3). The present analyses use data from Phases 1 and 2. In Phase 1, couples attended a lab session in which they provided informed consent and then completed several tasks, including a questionnaire measure of attachment orientations. The day immediately following the lab session, participants began Phase 2 and completed a 15-min series of questionnaires each day for 14 consecutive days. This survey included daily reports of relationship mindfulness and enacted positive and negative relationship behaviours. The survey was sent out at 4:00 p.m. each day and participants were asked to complete it by 11:59 p.m.. Each survey included an individual timestamped link which expired at midnight the following day to ensure participants completed each survey on the relevant day and completed only their own survey. The average number of daily surveys completed was high (range = 1–14, *M* = 12.96, *SD* = 2.01). Throughout both phases, partners were asked to complete tasks separately from one another. Ethical approval was obtained from the University of Edinburgh PPLS Research Ethics Committee (reference number: 15–1920/3).

#### 2.2.1. Attachment

In Phase 1, participants reported their dispositional attachment orientations using the Experiences in Close Relationships 12 (ECR-12) [44]. Six items measure attachment anxiety (e.g., “I worry a fair amount about losing my partner”) and six items measure attachment avoidance (e.g., “I feel comfortable depending on romantic partners”) rated on a 7-point scale (1 = *strongly disagree*, 7 = *strongly agree*). The ECR-12 was chosen because it provides a concise measure of attachment orientations while maintaining psychometric properties comparable to the original 36-item Experiences in Close Relationships Scale [13]. Scores were calculated by averaging responses across each subscale, with higher scores indicating greater anxiety and avoidance, respectively.

#### 2.2.2. Daily Relationship Mindfulness

In Phase 2, RM was measured daily using two items from the Relationship Mindfulness Measure (RMM) [7]. The RMM accounts for a significant portion of variance in relationship quality after controlling for general mindfulness [7,9]. Items were modified slightly to be suitable for the daily level by asking participants about their mindfulness “today”, rather than as a dispositional measure (e.g., “Today, when I was with my partner, I found myself saying or doing things without paying attention”) and were rated on a 6-point scale (1 = *almost never*, 6 = *almost always*). Only two items were used to increase efficiency and reduce the risk of participant fatigue and attrition during the diary period [45]. Item responses were reverse scored and averaged to create an RM score for each day of Phase 2, with higher scores indicating greater daily RM.

#### 2.2.3. Daily Relationship Behaviours

In Phase 2, participants reported their daily relationship behaviours across 24 items adapted from prior studies [46,47,48]. Participants responded to the stem “In the past 24 h, I…” and rated 12 negative relationship behaviours (e.g., “…said or did something that irritated my partner”) and 12 positive relationship behaviours (e.g., “…expressed gratitude for something my partner said or did”) on a 5-point scale (0 = *not at all*, 4 = *very much*). Items were averaged across each subscale to create relationship behaviour scores for each day of Phase 2, with higher scores indicating greater daily negative and positive relationship behaviours, respectively.

## 3. Results

Data analyses were guided by the Actor–Partner Interdependence Model (APIM), using multilevel modelling with necessary adjustments for indistinguishable dyadic data (i.e., nesting partners’ scores within a group of *N* = 2) [49]. The APIM allows for simultaneous testing of actor and partner effects and statistically controls for the mutual influence that exists between members of a dyad. We tested hypotheses with a moderated over time APIM [50], with separate models for attachment anxiety and avoidance. We ran two series of analyses, one where the outcome was same-day negative relationship behaviours and one where the outcome was same-day positive relationship behaviours. Following Garcia et al. [50], we explored all four possible two-way interactions: Actor Attachment × Actor RM, Actor Attachment × Partner RM, Partner Attachment × Actor RM, and Partner Attachment × Partner RM.

Effects of the interactions between attachment orientations and RM on next-day positive and negative behaviours were tested using lagged analyses with an ARH1 covariance structure to account for the fact that observations closer together in time are more similar to each other than observations further apart in time. Analyses involving lagged outcome variables controlled for the previous day’s report of the outcome [51]. Significant interactions were probed using simple slopes analyses. For ease of interpretation and to provide estimates of effect size, we standardised continuous predictors in all models. Descriptive statistics, reliability information, and correlations among study variables are provided in Table 1.

### 3.1. Associations with Attachment Anxiety

#### 3.1.1. Same-Day Negative Relationship Behaviours

As seen in the first outcome column of Table 2, a main effect of partner (but not actor) anxiety emerged; individuals with more anxiously attached partners reported enacting more same-day negative relationship behaviours. Main effects of actor and partner RM also emerged; an individual’s own as well as their partner’s relationship mindfulness predicted enacting fewer same-day negative relationship behaviours. Moreover, the Partner Anxiety × Actor RM interaction emerged (see Figure 1). Individuals with more anxious partners, but whose own daily RM was higher (vs. lower) on a given day reported fewer same-day negative behaviours, β (*SE*) = −0.10 (0.01), CI_95%_ = [−0.12, −0.08], *p* < 0.001. Individuals with less anxious partners, but whose own daily RM was higher (vs. lower) on a given day also reported fewer same-day negative behaviours, but the slope was not as steep, β (*SE*) = −0.05 (0.01), CI_95%_ = [−0.07, −0.03], *p* < 0.001. No other significant interactions emerged.

#### 3.1.2. Next-Day Negative Relationship Behaviours

As seen in the second outcome column of Table 2, no main or interactive effects of anxiety or RM emerged for next-day negative behaviours.

#### 3.1.3. Same-Day Positive Relationship Behaviours

As seen in the first outcome column of Table 3, no main effects of anxiety emerged; however, a main effect of actor (but not partner) RM emerged such that individuals’ own RM predicted enacting more same-day positive relationship behaviours. No significant interactions emerged.

#### 3.1.4. Next-Day Positive Relationship Behaviours

As seen in the second outcome column of Table 3, a main effect of actor prior-day RM emerged. When individuals reported lower (vs. higher) relationship mindfulness on a given day, they reported enacting more positive relationship behaviours the following day. Furthermore, the Partner Anxiety × Partner RM interaction emerged (see Figure 2). Individuals whose partner was more anxiously attached reported more next-day positive behaviours when their partner reported lower (vs. higher) prior-day RM, β (*SE*) = −0.06 (0.03), CI_95%_ = [−0.11, −0.007], *p* = 0.028. In contrast, individuals whose partners were less anxious reported equivalent levels of next-day positive behaviours regardless of their partners’ prior-day RM, β (*SE*) = 0.04 (0.03), CI_95%_ = [−0.01, 0.09], *p* = 0.154. No other main or interactive effects emerged.

### 3.2. Associations with Attachment Avoidance

#### 3.2.1. Same-Day Negative Relationship Behaviours

As seen in the first outcome column of Table 4, a main effect of actor (but not partner) avoidance emerged; more avoidantly attached individuals reported enacting more same-day negative relationship behaviours. Main effects of actor and partner RM also emerged; individuals’ own as well as their partners’ RM predicted enacting fewer same-day negative relationship behaviours. Moreover, the Actor Avoidance × Actor RM interaction revealed that less avoidant individuals reported enacting fewer negative behaviours on days when they had higher (vs. lower) RM, β (*SE*) = −0.09 (0.01), CI_95%_ = [−0.11, −0.07], *p* = < 0.001. More avoidant individuals also reported enacting fewer negative behaviours on days when they reported higher (vs. lower) RM, but the slope was less steep, β (*SE*) = −0.06 (0.01), CI_95%_ = [−0.08, −0.04], *p* < 0.001 (see Figure 3, Panel A).

The Actor Avoidance × Partner RM interaction emerged as well (see Figure 3, Panel B). More avoidantly attached individuals whose partners’ RM was higher (vs. lower) on a given day reported enacting fewer same-day negative relationship behaviours, β (*SE*) = −0.06 (.01), CI_95%_ = [−0.08, −0.04], *p* < 0.001. Less avoidant individuals reported equivalent levels of same-day negative behaviours regardless of their partners’ same-day RM, β (*SE*) = −0.02 (0.01), CI_95%_ = [−0.04, 0.005], *p* = 0.135.

Lastly, the Partner Avoidance × Partner RM interaction revealed that individuals whose partners were less avoidantly attached reported fewer same-day negative behaviours when their partners reported higher (vs. lower) same-day RM, β (*SE*) = −0.06 (.01), CI_95%_ = [−0.08, −0.03], *p* = < 0.001. Individuals with more avoidant partners also reported fewer same-day negative behaviours when their partners had higher (vs. lower) same-day RM, but the slope was less steep, β (*SE*) = −0.02 (.01), CI_95%_ = [−0.04, −0.002], *p* = 0.032 (see Figure 3, Panel C).

#### 3.2.2. Next-Day Negative Relationship Behaviours

As seen in the second outcome column of Table 4, a main effect of actor prior-day RM emerged. When individuals reported lower (vs. higher) RM on a given day, they reported enacting fewer negative behaviours the following day. No other main or interactive effects emerged.

#### 3.2.3. Same-Day Positive Relationship Behaviours

As seen in the first outcome column of Table 5, main effects of actor and partner attachment avoidance emerged. Individuals’ own and their partners’ attachment avoidance predicted enacting fewer same-day positive behaviours. Analyses also revealed a main effect of actor (but not partner) RM; on days when individuals reported higher (vs. lower) RM, they enacted more same-day positive relationship behaviours. Additionally, the Actor Avoidance × Actor RM interaction revealed that less avoidant individuals enacted more same-day positive behaviours when they had higher (vs. lower) RM on a given day, β (*SE*) = 0.10 (0.02), CI_95%_ = [0.05, 0.14], *p* < 0.001. More avoidant individuals reported equivalent same-day positive behaviours regardless of their same-day RM, β (*SE*) = 0.02(0.02), CI_95%_ = [−0.02, 0.06], *p* = 0.377 (see Figure 4, Panel A).

The Actor Avoidance × Partner RM interaction emerged as well (see Figure 4, Panel B). More avoidantly attached individuals whose partners’ RM was higher (vs. lower) on a given day reported enacting more same-day positive relationship behaviours, β (*SE*) = 0.07 (0.02), CI_95%_ = [0.02, 0.12], *p* = 0.003. Less avoidant individuals reported equivalent levels of same-day positive behaviours regardless of their partner’s same-day RM, β (*SE*) = −0.02 (0.02), CI_95%_ = [−0.06, 0.03], *p* = 0.493.

The Partner Avoidance × Actor RM interaction revealed that individuals whose partners were more avoidantly attached, but whose own same-day RM was higher (vs. lower) reported more same-day positive behaviours, β (*SE*) = 0.12 (0.02), CI_95%_ = [0.07, 0.16], *p* < 0.001. Those with less avoidant partners reported equivalent levels of same-day positive behaviours regardless of their own same-day RM, β (*SE*) = −0.001 (0.02), CI_95%_ = [−0.05, 0.05], *p* = 0.963 (see Figure 4, Panel C).

Lastly, the Partner Avoidance × Partner RM interaction emerged (see Figure 4, Panel D). Individuals whose partners were less avoidantly attached reported more same-day positive behaviours when their partners reported higher (vs. lower) same-day RM, β (*SE*) = 0.07 (0.02), CI_95%_ = [0.03, 0.12], *p* = 0.001. Individuals whose partners were more avoidant reported equivalent levels of same-day positive behaviours regardless of their partners’ same-day RM, β (*SE*) = −0.02 (0.02), CI_95%_ = [−0.06, 0.02], *p* = 0.357.

#### 3.2.4. Next-Day Positive Relationship Behaviours

As seen in the second outcome column of Table 5, a main effect of actor prior-day RM emerged; individuals who reported lower (vs. higher) RM on a given day reported more positive behaviours the following day. Furthermore, the Actor Avoidance × Partner RM interaction emerged such that more avoidantly attached individuals reported more next-day positive behaviours when their partners had lower (vs. higher) prior-day RM, β (*SE*) = −0.05 (0.03), CI_95%_ = [−0.11, −0.001], *p* = 0.047. Less avoidant individuals reported equivalent levels of next-day positive behaviours regardless of their partners’ prior-day RM, β (*SE*) = 0.03 (0.03), CI_95%_ = [−0.02, 0.08], *p* = 0.241 (see Figure 5). No other significant interactions emerged.

### 3.3. Auxiliary Analyses

In response to an anonymous peer reviewer, we reran our analyses controlling for sexual orientation, age, and relationship length. No effects of any control variable emerged in any model, and, importantly, our primary findings remained robust when controlling for these variables.

## 4. Discussion

The current study investigated how day-to-day RM might alter anxiously and avoidantly attached individuals’ same-day and next-day positive and negative relationship behaviours in a longitudinal dyadic study.

We found mixed support for CH1; attachment avoidance was associated with more negative and fewer positive relationship behaviours. These results align with prior theoretical and empirical literature regarding the detrimental consequences of avoidance on relationship outcomes, including behaviours [1,3,21,24]. However, we did not find this effect for attachment anxiety. This deviates from theory and substantial literature showing that the hyperactivating strategies associated with attachment anxiety often manifest in the form of more negative and fewer positive relationship behaviours (e.g., [1,3,15,19]). Since this finding has been so robust in previous literature, we assume this lack of an association may be a result of a peculiarity of this sample, or that other variables included in the model (e.g., RM) accounted for the variance.

We found consistent support for CH2; daily RM was associated with fewer same-day negative as well as more same-day positive behaviours. This is consistent with literature finding that both general mindfulness and RM are associated with positive relationship outcomes, including higher relationship quality, closeness, and more constructive responses to stress [9,31,33]. This also highlights the importance of day-to-day fluctuations in RM predicting behaviour, emphasising its importance in addition to trait measures of RM.

### 4.1. Attachment, RM, and Same-Day Behaviours

Interestingly, and in contrast to CH3, individuals’ own RM did not buffer their *own* insecure attachment. Although higher daily RM was associated with fewer negative behaviours for individuals both high and low in avoidance, the effect was stronger for those low (vs. high) in avoidance. Higher daily RM also predicted enacting more positive behaviours for individuals low in avoidance, but not those high in avoidance. This evidence thus supports a security-enhancing effect rather than an insecurity-buffering effect for the actor, as individuals low in avoidance appeared to reap greater benefits from their own RM. Unlike avoidance, and counter to CH3a, no Actor × Actor interactions emerged between attachment anxiety and RM. Shaver et al. [52] posit that individuals who grow up with caregivers who notice and articulate their own, and others’, thoughts, needs, and feelings develop a greater capacity for mindfulness. This may explain why fluctuations in outcomes on mindful versus non-mindful days may be larger for secure individuals; they may have a greater potential for mindfulness to begin with. In the same vein, while natural fluctuations in an individual’s RM across two weeks may not suffice to buffer their attachment insecurity, it is possible that trait RM may do so, as it may be indicative of more fundamental processes. The larger changes in RM which may occur in an intervention context may also serve to buffer insecurity [33].

Like Actor × Actor effects, the Partner × Partner effects found did not provide evidence of buffering. While higher partner RM was associated with enacting fewer negative behaviours for individuals with more and less avoidant partners, this effect was stronger for individuals with less avoidant partners. A similar pattern emerged when predicting same-day positive behaviours; individuals with less avoidant partners reported more positive behaviours on days when their partners reported high RM. This relation was nonsignificant for individuals with partners high in avoidance. Therefore, individuals’ own RM did not seem to buffer effects of their insecure attachment on their own or their partners’ behaviours. This ties in with the lack of Actor × Actor buffering effects; across the same partner, it appears that secure individuals benefit more from positive fluctuations in RM.

Although individuals’ own RM did not buffer their insecure attachment, we found cross-partner effects in which one partner’s RM buffered the other’s insecure attachment. People with both more and less anxious partners reported enacting fewer negative behaviours on days when they themselves reported higher (vs. lower) RM, but this effect was stronger for individuals with highly anxious partners. Our findings align with research suggesting that safe strategies, including soothing anxious concerns (e.g., through touch; [5] and diffusing hurt feelings [16]), can buffer anxiety, although in the current research, the benefits were limited to partner behaviours rather than those of the anxious individual themselves. Having higher RM may help the partners of anxious individuals engage more constructively in the face of hyperactivating strategies and negative emotions, allowing them to expend their resources in enacting safe strategies. However, there were no such effects for positive behaviours. In line with this, May et al. [39] found that mindfulness meditation practice was associated with a decrease in partner negative affect, but no increase in positive affect, showing that mindfulness may be more effective in reducing negative experiences than enhancing positive ones.

The partner buffering effects for both positive and negative behaviours were especially robust for attachment avoidance. For highly avoidant individuals, their partners’ higher (vs. lower) same-day RM was linked to the individuals themselves enacting fewer negative behaviours. Similarly, for more avoidant individuals, their partners’ higher (vs. lower) RM predicted more actor positive behaviours, while there was no association for less avoidant individuals. Likewise, for individuals with avoidant partners, their own higher (vs. lower) RM was associated with enacting more positive behaviours, while there was no association for individuals with less avoidant partners. This is consistent with the ASEM [4], which highlights the partner’s role in buffering attachment insecurity. Perhaps the empathy and understanding, increased awareness of automatic responses, or better emotion regulation associated with high RM days helped partners to tailor their responses to the beliefs and fears of avoidant individuals, thereby buffering the effects of partner avoidance on both their own and their partners’ behaviours. The findings for avoidance also dovetail with the research finding that avoidance is buffered by soft strategies. For example, partners being sensitive to avoidant individuals’ autonomy needs, acknowledging and appreciating their efforts, and engaging in rational and matter-of-fact, rather than emotional, phrasing appear to buffer avoidance [53,54]. The more consistent cross-partner buffering effects for avoidance than anxiety implies that partner RM may be more effective in helping partners enact soft rather than safe strategies. Alternatively, anxiety may be harder for partners to buffer than avoidance, particularly because it involves building up more positive working models of the self rather than others [3].

### 4.2. Attachment, RM, and Next-Day Behaviours

The pattern of effects on next-day relationship behaviours looked different from same-day behaviours. We also found no evidence that an individual’s own daily RM buffers the link between their own attachment anxiety or avoidance and their own next-day positive or negative behaviours. Unlike same-day behaviours, there was no security-enhancing effect either; neither secure nor insecure individuals appeared to benefit from prior-day RM. Contrary to EH1, as well as literature showing that RM is associated with positive relationship outcomes [7,9] and that general mindfulness may buffer long-term consequences of attachment insecurity [27], low prior-day RM was generally associated with better outcomes on the following day and there was no evidence of buffering. While this is further discussed below, it may mean that the benefits of RM do not carry over to the following day and that this may not work differentially for individuals who are securely versus insecurely attached when considering moderating effects of their own RM.

No next-day interactive Actor × Partner, Partner × Actor, or Partner × Partner effects emerged for negative behaviours for either attachment dimension. For positive behaviours, rather than RM buffering insecure attachment, it appears to predict worse next-day outcomes for avoidant individuals and for those with anxious partners. In the anxiety model predicting next-day positive behaviours, for individuals with highly anxious partners (but not for those with less anxious partners), their partners’ low (vs. high) daily RM actually predicted individuals enacting more next-day positive behaviours. It is possible RM as it was measured in the current study may have increased anxious individuals’ already strong hyper-focus on their partners and relationships [3]. Continually managing a partner’s insecurity and maintaining the relationship depletes an individual’s resources, eventually causing their efforts to wane or become inauthentic [4,55,56], and the effects of suppression of negative emotions on partner outcomes is attenuated in anxious individuals [57], indicating that lower RM may provide their partners with respite. If high RM in anxious individuals is related to their partners feeling more depleted in managing their insecurity, it would make sense that the partner would then lack the resources to enact fewer positive relationship behaviours the following day.

The avoidance effects, too, emerge differently for next-day versus same-day outcomes. More avoidant individuals reported more next-day positive behaviours when their partners reported lower (vs. higher) prior-day RM while there was no relation for less avoidant individuals. While avoidant individuals eschew intimacy and establish emotional distance from their partners [3], they may desire intimacy more than they may admit. This dovetails with Stanton et al.’s [6] findings that avoidant individuals reported low levels of enjoyment when engaging in an intimacy-promoting activity, but benefited nonetheless, reporting higher relationship quality, greater self-disclosure, and reduced avoidance over time. This may explain why avoidant individuals enact more positive behaviours the day after their partner exhibits low RM, perhaps with the aim of making their partner pay attention to and desire intimacy with them. Therefore, while high RM produces positive outcomes on the same day, *low* partner RM may be linked to better outcomes on the following day for individuals with anxious partners and for individuals who are avoidant.

### 4.3. Implications

Our findings have highlighted the important role a partner’s RM plays in buffering an individual’s insecure attachment, particularly for avoidant individuals. The identification of buffering effects of RM within cross-partner effects highlights the importance of examining attachment buffering as well as potential benefits of mindfulness in a dyadic context [8]. It also provides support for the propositions of the ASEM [4] regarding the role partners play in buffering attachment insecurity. Additionally, our findings emphasise the importance of examining day-to-day fluctuations in mindfulness, considering the differences found between same-day versus next-day outcomes. This supports Bishop’s [36] conceptualisation of mindfulness as a construct which can vary from day-to-day and is not static, as trait conceptions are. Moreover, our findings highlighting the patterns of daily RM fluctuation within a dyadic context have practical implications for couples’ therapy, demonstrating the need to emphasise the role of partner mindfulness in buffering attachment concerns. This may highlight the value of mindfulness-based training and therapy occurring in dyadic contexts, as it appears that partner RM may play a larger role in buffering insecure attachment than the individual’s own RM. Given the robust links between attachment insecurity and poorer physical and mental health [58], buffering insecurity has important implications for public health.

### 4.4. Limitations and Future Directions

Although the present research has several strengths, there are limitations worth noting. For example, the RM measure we used does not capture the full construct of mindfulness, as it focuses only on the attention and awareness component. This contrasts with measures such as the Five Factor Mindfulness Questionnaire (FFMQ) [59], which captures five facets: observing, describing, acting with awareness, nonreactivity, and nonjudging. The conception of mindfulness covered by the FFMQ covers the entire span of the original Buddhist conception of mindfulness. Attention and awareness alone may not buffer anxious attachment and, in fact, may not always be protective or constructive. For example, Baer et al. [59] found that the observing facet of the FFMQ (which is closely related to attention and awareness) was positively associated with psychological symptoms, disassociation, and thought suppression, which are negatively correlated with the other subscales. Furthermore, Pepping et al. [60] found a positive correlation between this observing facet and attachment anxiety. Together, these empirical findings suggest that assessing these associations using a relationship conceptualisation of a mindfulness scale, including facets of non-judgmental acceptance and nonreactivity while maintaining the interpersonal focus of the RMM (e.g., a partner-adapted version of the Interpersonal Mindfulness Scale [61]), would be a valuable future direction.

Another fruitful research direction is examining whether these associations extend to observable behaviours as well as self-reported behaviours. While relationship behaviours are predictive of global outcomes, including relationship quality, trust, and wellbeing [25,40,41], they may have different implications for global relationship outcomes if objectively observable. For example, Li and Chan [1] found that attachment anxiety and avoidance were more negatively associated with constructive interaction when this was measured using observational measures (vs. self-report measures), but more positively associated with destructive interactions when these were assessed using self-report (vs. observational) measures. Examining whether attachment orientations interact with RM to predict observer-rated behaviour may provide a more objective measure of this, avoiding social desirability concerns or the overestimation of own positive and underestimation of own negative behaviours. Additionally, with self-reported attachment, we could only measure conscious attachment strategies and this may not have picked up on attachment strategies individuals are unaware they engage in. Relatedly, while we have speculated that the cross-partner effects we found may be due to engagement in safe or soft strategies, we were unable to test this possibility in our data. Examining whether safe and soft strategies are mediating factors would be informative in explaining why partner RM buffers insecure attachment. Observing the strategies individuals engage in when their anxious or avoidant partners are distressed or withdraw would allow us to examine whether these strategies provide the pathway through which partner RM buffers individuals’ insecure attachment.

Furthermore, the current data examines ordinary day-to-day relationship behaviours. Investigating the role of RM in buffering attachment insecurity during times of chronic stress (e.g., illness of one partner or a close other, financial difficulties) would illuminate the role RM plays in buffering attachment insecurity over the course of long-lasting threatening events, which is likely to be more challenging than in day-to-day life. Because RM can be best understood as a construct which fluctuates even over short amounts of time [36], it may be that long-term stressful situations deplete an individual’s ability to apply RM and it may not be as effective a buffer in addressing a partner’s heightened insecurities at times of increased stress. Alternatively, RM may actually be an even more valuable buffer at these times because general mindfulness, which is positively associated with RM [7], may help individuals effectively deal with these external stressors. Generally, these challenges have been associated with lower relationship satisfaction and higher rates of relationship dissolution [62,63,64]; however, mindfulness may reduce this *stress spillover*, the impact of external stressors on relationship wellbeing [65]. Karremans et al. [8] argue that the increased stress and emotion regulation, awareness of otherwise automatic spillover, and increased support seeking associated with this awareness may protect the relationship in the face of external stressors which often deplete partners’ resources, including those which may be used to buffer insecurity.

While this study allowed us to examine the effects of day-to-day fluctuations in RM, the data are nonexperimental. Investigating this in an intervention context (e.g., a relationship-focused version of Mindfulness-Based Stress Reduction [33]) would allow us to establish true causal links. We could also draw clearer conclusions about how mindfulness training can be employed in couples’ therapy, particularly if, as we found, it is partner RM which buffers individuals’ insecure attachment.

Furthermore, while our data come from a representative community sample in the UK, the generalisability of our findings is somewhat limited by the fact that our sample was predominantly White. It would be valuable to attempt to replicate our findings in a more ethnically diverse sample.

## 5. Conclusions

In conclusion, the present study revealed daily RM as an important buffering factor for individuals who struggle with their relationships. While an individual’s own daily RM did not buffer the effects of their own insecure attachment on relationship behaviours, their partner’s daily RM did, particularly for avoidance. This emphasises the importance of examining the dyadic buffering effects of RM. Conversely, for next-day relationship behaviours, lower (vs. higher) prior-day RM was associated with better outcomes for avoidantly attached individuals and those with anxious partners, highlighting the importance of investigating the timeline of these processes. The most logical next steps for research in this area are to extend these findings by investigating how multiple facets of RM might buffer insecure attachment differently and examining whether partner RM increased through interventions can buffer insecure attachment over time. These findings provide more nuance to our understanding of an important process through which couples enjoy better relationship outcomes.

## Figures and Tables

**Figure 1 ijerph-17-07267-f001:**
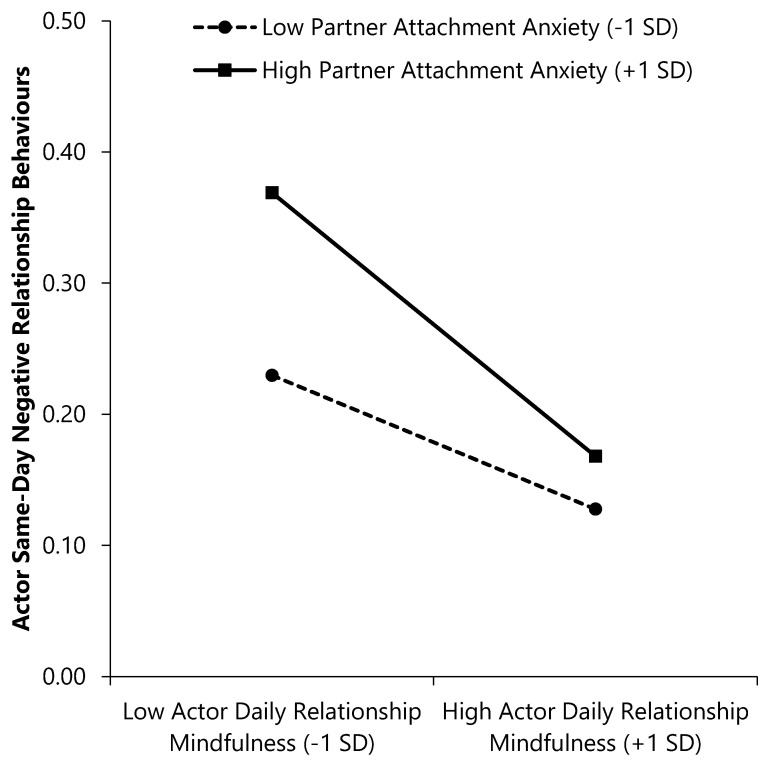
Interactions of attachment anxiety and daily relationship mindfulness with same-day negative relationship behaviours. Higher scores on continuous variables represent greater standing on the variable (e.g., higher daily relationship mindfulness (RM)).

**Figure 2 ijerph-17-07267-f002:**
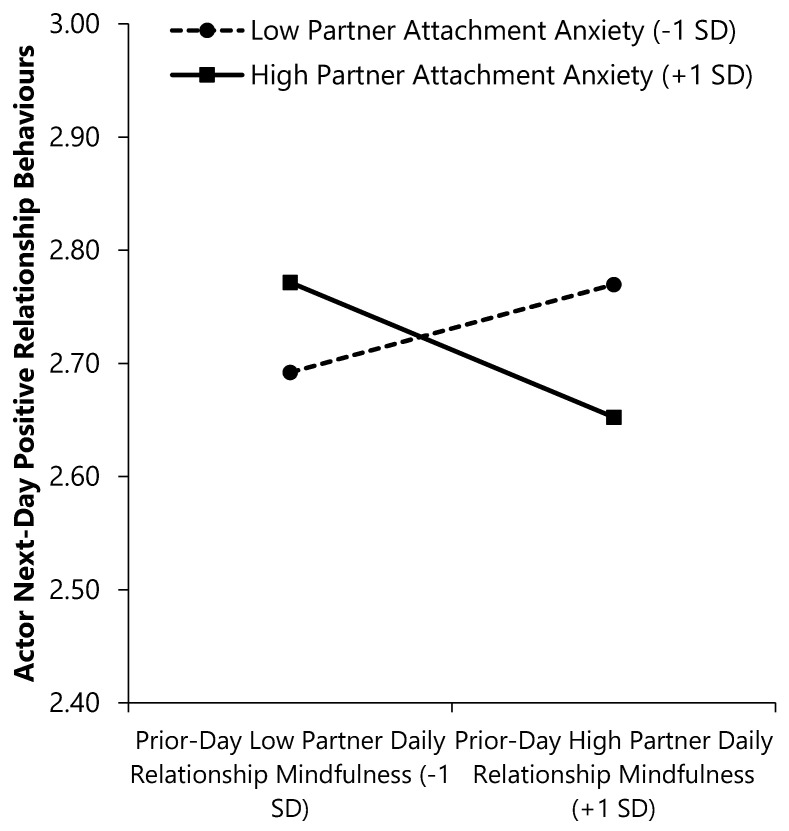
Interactions of attachment anxiety and prior-day relationship mindfulness with next-day positive relationship behaviours. Higher scores on continuous variables represent greater standing on the variable (e.g., higher daily RM).

**Figure 3 ijerph-17-07267-f003:**
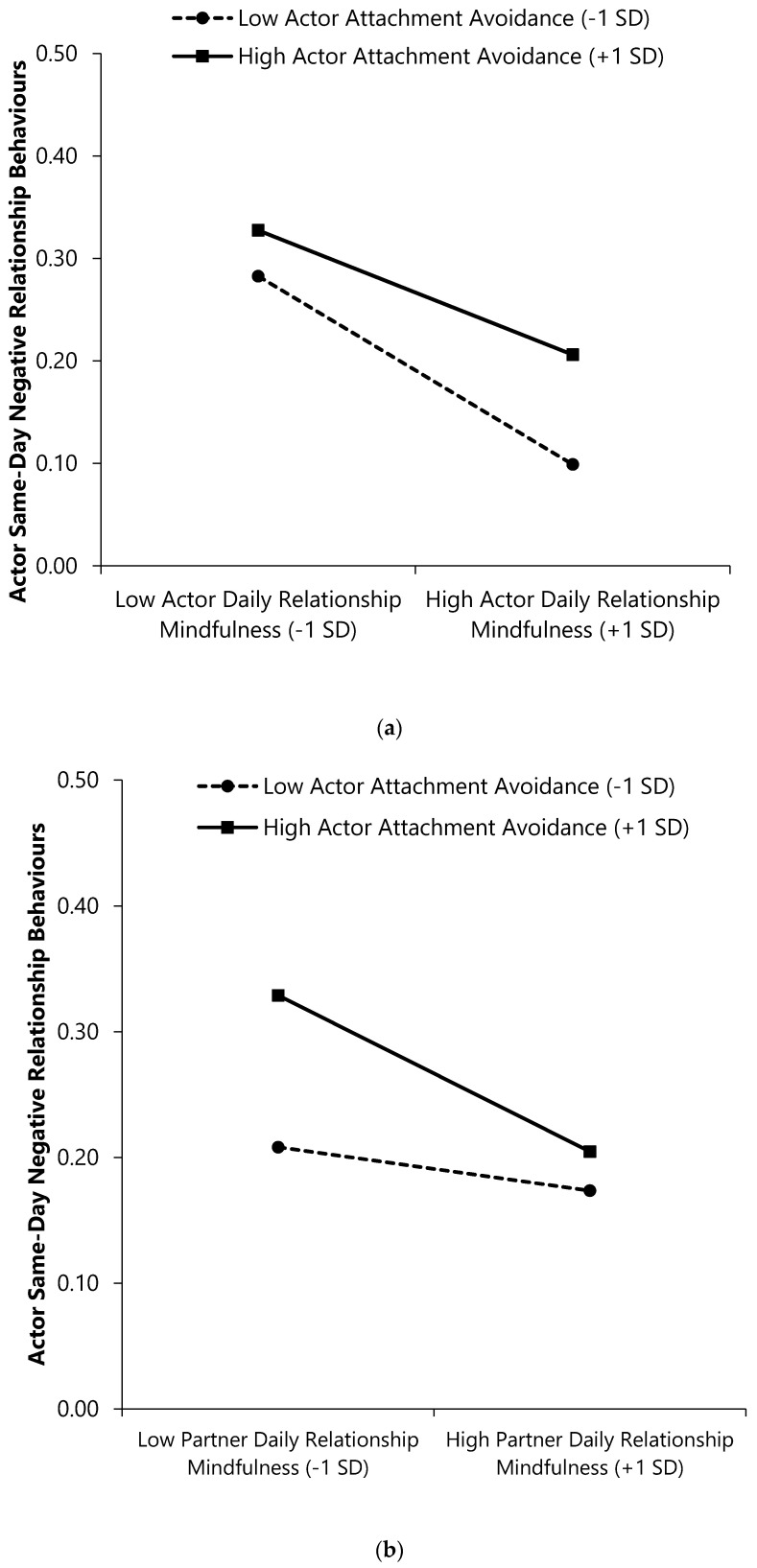
Interactions of attachment avoidance and daily relationship mindfulness with same-day negative relationship behaviours. (**a**) Actor × Actor interaction; (**b**) Actor × Partner interaction; (**c**) Partner × Partner interaction. Higher scores on continuous variables represent greater standing on the variable (e.g., higher daily RM).

**Figure 4 ijerph-17-07267-f004:**
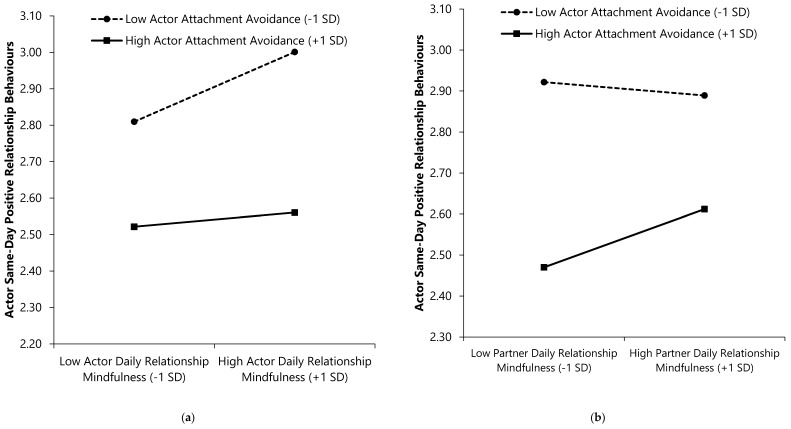
Interactions of attachment avoidance and daily relationship mindfulness with same-day positive relationship behaviours. (**a**) Actor × Actor interaction; (**b**) Actor × Partner interaction; (**c**) Partner × Actor interaction; (**d**) Partner × Partner interaction. Higher scores on continuous variables represent greater standing on the variable (e.g., higher daily RM).

**Figure 5 ijerph-17-07267-f005:**
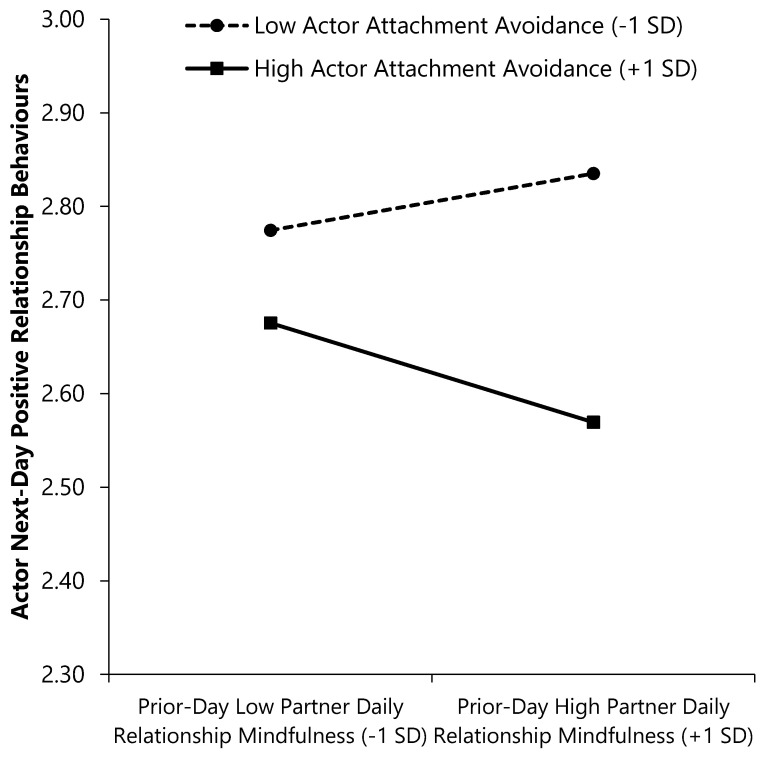
Interactions of attachment avoidance and prior-day relationship mindfulness with next-day positive relationship behaviours. Higher scores on continuous variables represent greater standing on the variable (e.g., higher daily RM).

**Table 1 ijerph-17-07267-t001:** Descriptive statistics, reliability information, and correlations among study variables.

		Descriptives and Reliability			Correlations				
Variable		Range	*M*(*SD*) or %	α	1	2	3	4	5
1	Attachment Anxiety	1.00–6.83	3.88 (1.29)	0.80	—				
2	Attachment Avoidance	1.00–5.83	2.27 (0.90)	0.78	−0.02	—			
3	Daily Relationship Mindfulness	1.00–6.00	5.33 (1.01)	0.91	−0.004	−0.10 ***	—		
4	Daily Negative Behaviours	0.00–3.17	0.22 (0.36)	0.82	0.03	0.13 ***	−0.21 ***	—	
5	Daily Positive Behaviours	0.00–4.00	2.73 (0.86)	0.91	0.02	0.24 ***	0.06 **	−0.20 ***	—

Note. *N* = 100 couples. Higher scores on continuous variables represent greater standing on the variable (e.g., higher daily relationship mindfulness). We present actor correlations. ** *p* < 0.01, *** *p* < 0.001.

**Table 2 ijerph-17-07267-t002:** Associations of attachment anxiety and daily relationship mindfulness with same-day and next-day negative relationship behaviours.

	Same-Day Negative Relationship Behaviours		Next-Day Negative Relationship Behaviours	
Predictor	β (*SE*)	CI_95%_	β (*SE*)	CI_95%_
A Attachment Anxiety	**0.01 (0.01)**	**[−0.02, 0.03]**	0.01 (0.01)	[−0.01, 0.03]
P Attachment Anxiety	**0.04 (0.01) *****	**[0.02, 0.07]**	0.03 (0.01) **	[0.01, 0.05]
A Same-Day RM	**−0.08 (0.01) *****	**[−0.09, −0.06]**	−0.07 (0.01) ***	[−0.09, −0.06]
P Same-Day RM	**−0.04 (0.01) *****	**[−0.05, −0.02]**	−0.03 (0.01) ***	[−0.05, −0.02]
A Anxiety × A Same-Day RM	**−0.004 (0.01)**	**[−0.02, 0.01]**	−0.02 (0.01)	[−0.03, 0.004]
A Anxiety × P Same-Day RM	**−0.01 (0.01)**	**[−0.03, 0.01]**	0.001 (0.01)	[−0.02, 0.02]
P Anxiety × A Same-Day RM	**−0.02 (0.01) ****	**[−0.04, −0.01]**	−0.01 (0.01)	[−0.03, 0.01]
P Anxiety × P Same-Day RM	**−0.01 (0.01)**	**[−0.03, 0.002]**	0.003 (0.01)	[−0.02, 0.02]
A Prior-Day RM			**0.02 (0.01)**	**[−0.001, 0.03]**
P Prior-Day RM			**0.01 (0.01)**	**[−0.01, 0.03]**
A Anxiety × A Prior-Day RM			**0.01 (0.01)**	**[−0.01, 0.03]**
A Anxiety × P Prior-Day RM			**−0.01 (0.01)**	**[−0.03, 0.01]**
P Anxiety × A Prior-Day RM			**0.01 (0.01)**	**[−0.01, 0.02]**
P Anxiety × P Prior-Day RM			**−0.01 (0.01)**	**[−0.02, 0.01]**
A Prior-Day Negative Behaviours			**0.10 (0.01) *****	**[0.09, 0.12]**

Note. N = 100 couples. A = actor; P = partner; RM = relationship mindfulness. Higher scores on continuous variables represent greater standing on the variable (e.g., higher daily RM). For clarity, the key effects in each model are in boldface. ** *p* < 0.01, *** *p* < 0.001.

**Table 3 ijerph-17-07267-t003:** Associations of attachment anxiety and daily relationship mindfulness with same-day and next-day positive relationship behaviours.

	Same-Day Positive Relationship Behaviours		Next-Day Positive Relationship Behaviours	
Predictor	β (*SE*)	CI_95%_	β (*SE*)	CI_95%_
A Attachment Anxiety	**0.02 (0.04)**	**[−0.06, 0.11]**	0.01 (0.03)	[−0.04, 0.06]
P Attachment Anxiety	**−0.01 (0.04)**	**[−0.10, 0.07]**	−0.01 (0.03)	[−0.06, 0.04]
A Same-Day RM	**0.06 (0.02)*****	**[0.03, 0.09]**	0.10 (0.02) ***	[0.06, 0.14]
P Same-Day RM	**0.02 (0.02)**	**[−0.01, 0.05]**	0.03 (0.02)	[−0.003, 0.07]
A Anxiety × A Same-Day RM	**0.01 (0.02)**	**[−0.03, 0.04]**	−0.01 (0.02)	[−0.04, 0.03]
A Anxiety × P Same-Day RM	**0.01 (0.02)**	**[−0.03, 0.04]**	0.002 (0.02)	[−0.04, 0.04]
P Anxiety × A Same-Day RM	**−0.0001 (0.02)**	**[−0.03, 0.03]**	0.02 (0.02)	[−0.02, 0.05]
P Anxiety × P Same-Day RM	**0.01 (0.02)**	**[−0.03, 0.04]**	0.03 (0.02)	[−0.01, 0.07]
A Prior-Day RM			**−0.050 (0.02) ****	**[−0.09, −0.02]**
P Prior-Day RM			**−0.01 (0.02)**	**[−0.05, 0.03]**
A Anxiety × A Prior-Day RM			**−0.01 (0.02)**	**[−0.04, 0.03]**
A Anxiety × P Prior-Day RM			**−0.01 (0.02)**	**[−0.05, 0.03]**
P Anxiety × A Prior-Day RM			**−0.03 (0.02)**	**[−0.07, 0.01]**
P Anxiety × P Prior-Day RM			**−0.05 (0.02) ***	**[−0.09, −0.01]**
A Prior-Day Positive Behaviours			**0.41 (0.02) *****	**[0.38, 0.44]**

Note. N = 100 couples. A = actor; P = partner; RM = relationship mindfulness. Higher scores on continuous variables represent greater standing on the variable (e.g., higher daily RM). For clarity, the key effects in each model are in boldface. * *p* < 0.05, ** *p* < 0.01, *** *p* < 0.001.

**Table 4 ijerph-17-07267-t004:** Associations of attachment avoidance and daily relationship mindfulness with same-day and next-day negative relationship behaviours. Note. N = 100 couples. A = actor; P = partner; RM = relationship mindfulness. Higher scores on continuous variables represent greater standing on the variable (e.g., higher daily RM). For clarity, the key effects in each model are in boldface. * *p* < 0.05, ** *p* < 0.01, *** *p* < 0.001.

	Same-Day Negative Relationship Behaviours		Next-Day Negative Relationship Behaviours	
Predictor	β (*SE*)	CI_95%_	β (*SE*)	CI_95%_
A Attachment Avoidance	**0.04 (0.01) ****	**[0.01, 0.06]**	0.02 (0.01) *	[0.004, 0.04]
P Attachment Avoidance	**−0.01 (0.01)**	**[−0.03, 0.02]**	−0.001(0.01)	[−0.02, 0.02]
A Same-Day RM	**−0.08 (0.01) *****	**[−0.09, −0.06]**	−0.08 (0.01) ***	[−0.09. −0.06]
P Same-Day RM	**−0.04 (0.01) *****	**[−0.06, −0.02]**	−0.03 (0.01) ***	[−0.05, −0.01]
A Avoidance × A Same-Day RM	**0.02 (0.01) ***	**[0.001, 0.03]**	0.001 (0.01)	[−0.01, 0.03]
A Avoidance × P Same-Day RM	**−0.02 (0.01) ****	**[−0.04, −0.01]**	−0.01 (0.01)	[−0.02, 0.01]
P Avoidance × A Same-Day RM	**−0.01 (0.01)**	**[−0.02, 0.01]**	−0.01 (0.01)	[−0.03, 0.01]
P Avoidance × P Same-Day RM	**0.02 (0.01) ***	**[0.002, 0.03]**	0.01 (0.01)	[−0.01, 0.03]
A Prior-Day RM			**0.02 (0.01) ***	**[0.001, 0.04]**
P Prior-Day RM			**0.01 (0.01)**	**[−0.01, 0.03]**
A Avoidance × A Prior-Day RM			**0.01 (0.01)**	**[−0.01, 0.02]**
A Avoidance × P Prior-Day RM			**−0.003 (0.01)**	**[−0.02, 0.02]**
P Avoidance × A Prior-Day RM			**0.004 (0.01)**	**[−0.01, 0.02]**
P Avoidance × P Prior-Day RM			**0.01 (0.01)**	**[−0.01, 0.02]**
A Prior-Day NegativeBehaviours			**0.10 (0.01) *****	**[0.09, 0.12]**

**Table 5 ijerph-17-07267-t005:** Associations of attachment avoidance and daily relationship mindfulness with same-day and next-day positive relationship behaviours. Note. N = 100 couples. A = actor; P = partner; RM = relationship mindfulness. Higher scores on continuous variables represent greater standing on the variable (e.g., higher daily RM). For clarity, they key effects in each model are in boldface. * *p* < 0.05, ** *p* < 0.01, *** *p* < 0.001.

	Same-Day Positive Relationship Behaviours		Next-Day Positive Relationship Behaviours	
Predictor	β (*SE*)	CI_95%_	β (*SE*)	CI_95%_
A Attachment Avoidance	**−0.18 (0.04)*****	**[−0.26, −0.11]**	−0.09 (0.02) ***	[−0.14, −0.04]
P Attachment Avoidance	**−0.11 (0.04)****	**[−0.18, −0.03]**	−0.06 (.02) **	[−0.11, −0.02]
A Same-Day RM	**0.06 (0.02)*****	**[0.03, 0.09]**	0.10 (0.02) ***	[0.06, 0.14]
P Same-Day RM	**0.03 (0.02)**	**[−0.01, 0.06]**	0.03 (0.02)	[−0.002, 0.07]
A Avoidance × A Same-Day RM	**−0.04 (0.02)***	**[−0.07, −0.01]**	−0.04 (0.02) *	[−0.07, −0.01]
A Avoidance × P Same-Day RM	**0.04 (0.02)***	**[0.01, 0.08]**	0.06 (0.02) **	[0.03, 0.10]
P Avoidance × A Same-Day RM	**0.06 (0.02)****	**[0.02, 0.09]**	0.06 (0.02) **	[0.02, 0.10]
P Avoidance × P Same-Day RM	**−0.05 (0.02)****	**[−0.08, −0.02]**	0.03 (0.02)	[−0.06, 0.003]
A Prior-Day RM			**−0.06 (0.02) ****	**[−0.09, −0.02]**
P Prior-Day RM			**−0.01 (0.02)**	**[−0.05, 0.02]**
A Avoidance × A Prior-Day RM			**0.02 (0.02)**	**[−0.02, 0.05]**
A Avoidance × P Prior-Day RM			**−0.04 (0.02) ***	**[−0.08, −0.004]**
P Avoidance × A Prior-Day RM			**−0.01 (0.02)**	**[−0.05, 0.02]**
P Avoidance × P Prior-Day RM			**−0.01 (0.02)**	**[−0.04, 0.02]**
A Prior-Day Positive Behaviours			**0.40** (**0.02**) *******	**[0.37, 0.43]**

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
