# Peer review of "Partners’ Relationship Mindfulness Promotes Better Daily Relationship Behaviours for Insecurely Attached Individuals"

_ijerph, 2020, doi:10.3390/ijerph17197267_

Round 1
Reviewer 1 Report
I have three major concerns for this manuscript - (1) methodological and theoretical generalizability: how can the authors come to a generalizable conclusion with selected sample; (2) Compatibility with the scope of IJERPG: the authors need to explicitly explain why this research match to the main topics of IJERPH including issues and topics concerning environment, public health, public policy etc.; (3) Structural adjustment: Chapter 1 is too long. 1.1-1.4 can be restructured as an independent chapter of theory and hypothesis.
Author Response
We have enclosed a revised version of the manuscript entitled, “Partners’ Relationship Mindfulness Promotes Better Daily Relationship Behaviours for Insecurely Attached Individuals” (ijerph-932315). We were grateful for the reviewers’ comments and have taken them to heart as we revised the manuscript.
In the following sections, we address the reviewers’ feedback. When addressing concerns, we quote your comments in underlined type. Page and line numbers listed in this letter pertain to the tracked version of the revised manuscript, which contains the new and revised material highlighted in yellow for ease.
In your comments, you highlighted three concerns. We address each in turn.
- Methodological and theoretical generalizability: how can the authors come to a generalizable conclusion with selected sample
We appreciate the highlighting of the importance of generalisability. Our sample was a representative community sample from the UK and we therefore have reasonable confidence in the generalisability of our findings. However, it is certainly true that our sample was predominantly white, which we have now discussed on lines 585 to 587 of our discussion section.
- Compatibility with the scope of IJERPG: the authors need to explicitly explain why this research match to the main topics of IJERPH including issues and topics concerning environment, public health, public policy etc.
This is an important point which we have addressed on line 166 of our introduction and lines 525 to 527 of our discussion section where we discuss the implications of attachment orientations (and hence the importance of buffering insecure attachment) for public health.
- Structural adjustment: Chapter 1 is too long. 1.1-1.4 can be restructured as an independent chapter of theory and hypothesis.
We appreciated this suggestion and have endeavoured to reduce the length of the introduction section. However, we had to balance responding to this concern with Reviewer 2’s request for more information to be provided in several places in the introduction. Overall, the length of Chapter 1 is fairly reduced.
Once again, we thank you for your helpful feedback. If you have any questions about the changes we have made, or if you would like to see further changes made prior to making a final decision about the manuscript, please let us know. We look forward to hearing your suggestions for improving the paper further.
We close by noting that the research discussed in this manuscript has not been published previously and is not under consideration for publication at any other journal. Publication of this manuscript is approved by all authors.
Thank you very much for your consideration.
Sincerely,
The Authors
Reviewer 2 Report
Article title: Partners’ Relationship Mindfulness Promotes Better Daily Relationship Behaviours for Insecurely Attached Individuals
Drawing on the literature that highlights the buffering of insecure attachment styles by specific factors the authors aim to investigate the effect of Relationship Mindfulness (RM) in improving 15 day-to-day relationship behaviors for anxious and avoidant individuals. The authors found that an individual’s own daily RM did not buffer the effects of their own insecure attachment on same-day relationship behaviours, but that their partner’s daily RM did in the case of attachment avoidance. As far as it concerns next-day relationship behaviours the authors found that lower (vs. higher) prior-day RM was associated with higher positive partner behaviours on the following day for avoidant individuals and those with anxious partners. The authors conclude that findings support the Attachment Security Enhancement Model. I think that the article is of particular relevance with respect to the attempt to frame relationship dynamics through an attachment lens and integrating constructs emerging from the fields of studies inherent Mindfulness. Anyway, I think that some points in the manuscript should be revised in order to improve its quality and readability. My suggestions for revisions are reported following.
Introduction
(line 29) Please explain what is meant by attachment orientation.
(line 32) Please. provide some example of relationship negative outcomes to which anxiety and avoidance are associated to
(line 33) The authors make reference to insecure attachment but the construct has not been explained before, please provide a brief description although it will be further discussed below.
(lines 48-50) Please further explain what we mean with securely attached individuals (rather than from scoring low on anxiety and avoidance). What does this mean in terms of relationship? I suggest also to further address to what attachment insecurity means. For example the authors could further explain how this aspects could prevent individuals from relying on the other as a secure base in moments of need.
(Lines 91-92) please specify that the Attachment Security Enhancement Model will be referred to as ASEM following in the manuscript
Materials &Methods
(line 220) in paragraph 2.2. insert a line specifying that participants were asked to sign an informed consent.
In socio-demographic data the authors report information inherent sexual orientation, age and length of relationship, were these variables controlled during statistical analyses? (it is likely that RM and attachment categories reached a different level of functioning and interconnection between two individuals that have been together for a long time, with respect to individuals that recently engaged together). If so please report that, otherwise provide rationale for this choice in the methods section and further address this aspect in the limitations section.
The tables are at times difficult to read given the large amount of information provided. I suggest the possibility to split them in two parts one dedicated to the actor results and the other one to the partner results, so to make them more easily readable.
Limitation section
Please further address issues inherent limitations due to the instruments adopted, for example the fact of using self reported measures for measuring attachment (and subsequently the possibility to measure only conscious strategies), and the fact that RM was measured using only two items of another questionnaire, which probably did not capture
Author Response
We have enclosed a revised version of the manuscript entitled, “Partners’ Relationship Mindfulness Promotes Better Daily Relationship Behaviours for Insecurely Attached Individuals” (ijerph-932315). We were grateful for the reviewers’ comments and have taken them to heart as we revised the manuscript.
In the following sections, we address the reviewers’ feedback. When addressing concerns, we quote your comments in underlined type. Page and line numbers listed in this letter pertain to the tracked version of the revised manuscript, which contains the new and revised material highlighted in yellow for ease.
In your comments, you highlighted nine concerns. We address each in turn.
- (line 29) Please explain what is meant by attachment orientation.
We appreciated this suggestion for providing context and have now added an explanation of attachment orientation on lines 30 to 32.
- (line 32) Please. provide some example of relationship negative outcomes to which anxiety and avoidance are associated to
We appreciated this suggestion and have provided some examples of general negative relationship outcomes which attachment anxiety and avoidance are associated with on lines 35 to 36.
- (line 33) The authors make reference to insecure attachment but the construct has not been explained before, please provide a brief description although it will be further discussed below.
We were happy to provide a definition of insecure attachment on lines 37 to 38.
- (lines 48-50) Please further explain what we mean with securely attached individuals (rather than from scoring low on anxiety and avoidance). What does this mean in terms of relationship? I suggest also to further address to what attachment insecurity means. For example the authors could further explain how this aspects could prevent individuals from relying on the other as a secure base in moments of need.
We appreciated the suggestion of providing context for what secure attachment refers to and means in a relationship context and have inserted an explanation of this on lines 54 to 58 and 62 to 66.
- (Lines 91-92) please specify that the Attachment Security Enhancement Model will be referred to as ASEM following in the manuscript
This is a very good point and we have noted this on line 95.
- (line 220) in paragraph 2.2. insert a line specifying that participants were asked to sign an informed consent.
We appreciated this suggestion and have specified this on line 218.
- In socio-demographic data the authors report information inherent sexual orientation, age and length of relationship, were these variables controlled during statistical analyses? (it is likely that RM and attachment categories reached a different level of functioning and interconnection between two individuals that have been together for a long time, with respect to individuals that recently engaged together). If so please report that, otherwise provide rationale for this choice in the methods section and further address this aspect in the limitations section.
This was a valuable suggestion which we were happy to address in our manuscript. After controlling for sexual orientation, age, and length of relationship, all our findings remained robust. There were no effects of any control variable in any model. This is noted on lines 374 to 377.
- The tables are at times difficult to read given the large amount of information provided. I suggest the possibility to split them in two parts one dedicated to the actor results and the other one to the partner results, so to make them more easily readable.
We appreciate that the tables in this manuscript contain a large amount of information and carefully considered the suggestion of splitting them into two parts to make them more readable. However, we elected to keep the current tables as they are for two reasons Firstly, because a substantial portion of our tables is dedicated to actor × partner interactions, it would be difficult to categorise sections of the tables as “actor” or “partner” results. In addition to this, formatting tables in this fashion conforms to convention in the relationships literature (e.g., Dobson et al., 2020).
- Please further address issues inherent limitations due to the instruments adopted, for example the fact of using self reported measures for measuring attachment (and subsequently the possibility to measure only conscious strategies), and the fact that RM was measured using only two items of another questionnaire, which probably did not capture
We were happy to address the limitations of only being able to measure the use of conscious strategies in the present research on lines 555 to 557 of the discussion. In reference to the limitations of using only two items from the original relationship mindfulness measure, we did so in order to increase efficiency and to minimise the risk of participant fatigue and attrition. Moreover, the two items demonstrated excellent internal reliability (α = .91), reliability which is comparable to the original measure. Therefore, we feel confident that we captured daily relationship mindfulness with our items.
Once again, we thank you for your helpful feedback. If you have any questions about the changes we have made, or if you would like to see further changes made prior to making a final decision about the manuscript, please let us know. We look forward to hearing your suggestions for improving the paper further.
We close by noting that the research discussed in this manuscript has not been published previously and is not under consideration for publication at any other journal. Publication of this manuscript is approved by all authors.
Thank you very much for your consideration.
Sincerely,
The Authors
Round 2
Reviewer 2 Report
Dear authors,
thank you for precisely addressing to all of the queries. I think the manuscript is more easily readable and has definitely improved now.